# Reduced SLIT2 is Associated with Increased Cell Proliferation and Arsenic Trioxide Resistance in Acute Promyelocytic Leukemia

**DOI:** 10.3390/cancers12113134

**Published:** 2020-10-27

**Authors:** Isabel Weinhäuser, Diego A. Pereira-Martins, Cesar Ortiz, Douglas R. Silveira, Luíse A. A. Simões, Thiago M. Bianco, Cleide L. Araujo, Luisa C. Koury, Raul A. M. Melo, Rosane I. Bittencourt, Katia Pagnano, Ricardo Pasquini, Elenaide C. Nunes, Evandro M. Fagundes, Ana B. Gloria, Fábio Kerbauy, Maria de Lourdes Chauffaille, Armand Keating, Martin S. Tallman, Raul C. Ribeiro, Richard Dillon, Arnold Ganser, Bob Löwenberg, Peter Valk, Francesco Lo-Coco, Miguel A. Sanz, Nancy Berliner, Emanuele Ammatuna, Antonio R. Lucena-Araujo, Jan Jacob Schuringa, Eduardo M. Rego

**Affiliations:** 1Department of Internal Medicine, Medical School of Ribeirao Preto, University of Sao Paulo, Ribeirao Preto 14048-900, Brazil; i.weinhauser@umcg.nl (I.W.); d.a.pereira.martins@umcg.nl (D.A.P.-M.); ortizr@usp.br (C.O.); tmbianco@usp.br (T.M.B.); luisa.koury.lk@gmail.com (L.C.K.); 2Department of Medical Images, Hematology, and Clinical Oncology, Medical School of Ribeirao Preto, University of Sao Paulo, Ribeirao Preto 14048-900, Brazil; 3Center for Cell Based Therapy, University of São Paulo, Ribeirao Preto 14051-060, Brazil; luise.simoes@fm.usp.br (L.A.A.S.); cleide@hemocentro.fmrp.usp.br (C.L.A.); 4Department of Experimental Hematology, Cancer Research Centre Groningen, University Medical Centre Groningen, University of Groningen, 9700 RB Groningen, The Netherlands; e.ammatuna@umcg.nl; 5Hematology Division, LIM31, Faculdade de Medicina, University of Sao Paulo, Sao Paulo 05403-000, Brazil; douglas.rafaele@usp.br; 6Department of Hematology, AC Camargo Cancer Center, Sao Paulo 01525-001, Brazil; 7Department of Internal Medicine, University of Pernambuco, Recife 50100-130, Brazil; raul.melo@outlook.com; 8Hematology Division, Federal University of Rio Grande do Sul, Porto Alegre 96.200-190, Brazil; rbittencourt@hcpa.edu.br; 9Hematology and Hemotherapy Center, University of Campinas, Campinas 13083-878, Brazil; kborgia@unicamp.br; 10Hematology Division, Federal University of Parana, Curitiba 13083-878, Brazil; pasquini@hc.ufpr.br (R.P.); elenaide@gmail.com (E.C.N.); 11Hematology Division, Federal University of Minas Gerais, Belo Horizonte 30130-100, Brazil; evandro.fagundes@oncoclinicas.com (E.M.F.); anabeatrizfirmatog@gmail.com (A.B.G.); 12Hematology Division, Federal University of Sao Paulo, Sao Paulo 04023-062, Brazil; fkerbauy@gmail.com (F.K.); mlourdes.chauffaille@grupofleury.com.br (M.d.L.C.); 13Princess Margaret Cancer Centre, Toronto, ON M5G 2C1, Canada; armand.keating@uhn.on.ca; 14Leukemia Service, Memorial Sloan-Kettering Cancer Center/Weill Cornell Medical College, New York, NY 10065, USA; tallmanm@mskcc.org; 15Department of Oncology, St. Jude Children’s Research Hospital, Memphis, TN 38105, USA; Raul.Ribeiro@STJUDE.ORG; 16Department of Medical and Molecular Genetics, King’s College London School of Medicine, London WC2R 2LS, UK; richard.dillon@kcl.ac.uk; 17Department of Hematology, Hemostasis, Oncology, and Stem Cell Transplantation, Hannover Medical School, 30625 Hannover, Germany; Ganser.Arnold@mh-hannover.de; 18Department of Hematology, Erasmus University Medical Center, 3015 GD Rotterdam, The Netherlands; b.lowenberg@erasmusmc.nl (B.L.); p.valk@erasmusmc.nl (P.V.); 19Department of Biopathology, University Tor Vergata, 00133 Rome, Italy; francesco.lo.coco@uniroma2.it; 20Santa Lucia Foundation, 00179 Rome, Italy; 21Department of Hematology, Hospital Universitari I Politècnic La Fe, 46-009 Valencia, Spain; Miguel.Sanz@uv.es; 22CIBERONC, Instituto Carlos III, 28029 Madrid, Spain; 23Department of Medicine, Brigham and Women’s Hospital, Harvard Medical School, Boston, MA 02115, USA; nberliner@bwh.harvard.edu; 24Department of Genetics, Federal University of Pernambuco, Recife 50670-901, Brazil; araujoarl@gmail.com

**Keywords:** acute promyelocytic leukemia, SLIT2, treatment outcomes, ATRA

## Abstract

**Simple Summary:**

In solid tumors, the altered expression of embryonic genes such as the SLIT-ROBO family has been associated with poor prognosis, while little is known about their role in acute myeloid leukemia (AML). Previous studies reported frequent hypermethylation of SLIT2 mediated by the methyltransferase enzyme EZH2 and more recently the PML protein, which are commonly found to be aberrantly expressed in AML. Here, we aim to assess retrospectively the clinical relevance of the SLIT2 gene in acute promyelocytic leukemia, a homogenous subtype of AML. We demonstrated that reduced SLIT2 expression was associated with high leukocyte counts and reduced overall survival in different APL cohorts. STLI2 treatment decreased APL growth, while SLIT2 knockdown accelerated cell cycle progression and proliferation. Finally, reduced expression of SLIT2 in murine APL blasts resulted in fatal leukemia associated with increased leukocyte counts in vivo. These findings demonstrate that SLIT2 can be considered as a prognostic marker in APL, and a potential candidate for clinical studies of a more heterogeneous disease, such as AML.

**Abstract:**

The SLIT-ROBO axis plays an important role in normal stem-cell biology, with possible repercussions on cancer stem cell emergence. Although the Promyelocytic Leukemia (PML) protein can regulate *SLIT2* expression in the central nervous system, little is known about SLIT2 in acute promyelocytic leukemia. Hence, we aimed to investigate the levels of SLIT2 in acute promyelocytic leukemia (APL) and assess its biological activity in vitro and in vivo. Our analysis indicated that blasts with *SLIT2^high^* transcript levels were associated with cell cycle arrest, while *SLIT2^low^* APL blasts displayed a more stem-cell like phenotype. In a retrospective analysis using a cohort of patients treated with all-trans retinoic acid (ATRA) and anthracyclines, high *SLIT2* expression was correlated with reduced leukocyte count (*p* = 0.024), and independently associated with improved overall survival (hazard ratio: 0.94; 95% confidence interval: 0.92–0.97; *p* < 0.001). Functionally, *SLIT2*-knockdown in primary APL blasts and cell lines led to increased cell proliferation and resistance to arsenic trioxide induced apoptosis. Finally, in vivo transplant of Slit2-silenced primary APL blasts promoted increased leukocyte count (*p* = 0.001) and decreased overall survival (*p* = 0.002) compared with the control. In summary, our data highlight the tumor suppressive function of SLIT2 in APL and its deteriorating effects on disease progression when downregulated.

## 1. Introduction

In the event of tumorigenesis, tumor stem cells are a small, but essential subpopulation, which retain the function of self-renewal necessary to generate and sustain malignant tissue formation [1]. The resemblance between the structural process of tumor formation and embryonic development led to the assumption that embryonic molecular cues may be implicated in cancer progression [2].

One such candidate is the SLIT-ROBO family. SLIT is a glycoprotein mainly expressed in the central nervous system (CNS) consisting of three homologs: SLIT1, SLIT2, and SLIT3, which can bind to their respective roundabout (ROBO) receptors: ROBO1, ROBO2, ROBO3, and ROBO4 [2]. The SLIT-ROBO axis was described to regulate axon guidance along the midline during the development of the central nervous system (CNS) [3]. Later, the SLIT-ROBO axis was implicated in a wide range of biological processes such as organogenesis, angiogenesis, cell proliferation, differentiation, stem cell regulation, and migration [4,5]. In the context of cancer, SLIT-ROBO has been attributed to both tumor suppressive and pro-oncogenic functions, indicating tissue-specific activities. When acting as a tumor suppressor, the promotor site of SLIT-ROBO is frequently hypermethylated [2]. While SLIT-ROBO has been extensively studied in solid tumors, its function in leukemia, more precisely acute promyelocytic leukemia (APL), is still unclear [6].

APL is a very homogenous disease characterized by the PML-RARA translocation with exceptionally good prognosis when treated with all-trans retinoic acid (ATRA) based therapy in combination with chemotherapy or arsenic trioxide (ATO) [7,8]. Despite the revolution of ATRA and ATO, high risk patients (defined by a white blood cell (WBC) count above 10 × 10^9^/L) [8] are still a challenge in the clinic. Yet, it remains unclear which biological process beyond the PML-RARA differentiation block causes such an aggressive phenotype. On this note, a recent study on glioblastoma established a novel link between the SLIT1/2 protein and PML nuclear body formation [9]. Amodeo et al. reported that the PML driven suppression of SLIT2 is mediated by the polycomb repressive complex 2 (PRC2) promoting histone 3 lysine 27 tri-methylation (H3K27me3). As a result, the authors demonstrated an upregulation of SLIT2 and SLIT1 upon PML deficiency under physiological and oncogenic conditions, respectively.

Given these insights, we sought to investigate the function SLIT/PML axis in APL, where the formation of PML nuclear body is abrogated [10]. In our study, low *SLIT2* expression was associated with high WBC count and poor clinical outcome. In vitro the knockdown of *SLIT2* significantly increased APL cell proliferation, conferred ATO resistance, and led to decreased overall survival in vivo.

## 2. Results

### 2.1. Low SLIT2 Transcript Levels Predict Lower Overall Survival in APL Characterized by a More Aggressive Course of Disease Progression

To determine the transcript levels of SLIT2, ROBO1, and ROBO4 in healthy bone marrow (BM) hematopoietic stem-progenitors cells (HSPCs; defined by Hematopoietic Stem Cells [HSC], Common myeloid progenitors [CMP], and Granulocyte-macrophage progenitors [GMP] cells; *n* = 18), healthy promyelocytes (*n* = 6), and blasts from APL patients (*n* = 54), we used the public databank BloodSpot [11]. In general, APL patients displayed increased expression of SLIT2 and reduced levels of SLIT1 and ROBO1, 3, and 4 compared with healthy HSPCs and promyelocytes (*p* < 0.001; Appendix A).

To evaluate the impact of SLIT2 levels on clinical and laboratory characteristics, we dichotomized our APL patients from the International Consortium of Acute Promyelocytic Leukemia (IC-APL) cohort [12] (IC-APL, *n* = 94) into low and high SLIT2 expression using the first quartile (p25) value. Baseline characteristics were similar between patients (Table 1), except for higher white blood cell (WBC) counts in patients assigned to the low SLIT2 group (*p* = 0.024). Additionally, SLIT2 transcript levels negatively correlated with white blood cell (WBC) counts (Pearson coefficient, r: −0.2335; *p* = 0.0286). Similar correlations were observed when we analyzed APL samples from public datasets (Figure 1A).

The median follow-up among survivals in the IC-APL cohort was 40 months (95% confidence interval (CI): 36–43 months). Using the categorical classification for SLIT2 expression, patients with low SLIT2 expression presented lower 5-year overall survival (OS) (68%, 95% CI: 45–83%) compared with those with high SLIT2 expression (84%, 95% CI: 73–91%; *p* = 0.074, Figure 1B). Furthermore, the best-fitted multivariate Cox proportional hazards model for OS, including age (continuous variable), leukocyte counts (continuous variable), and SLIT2 transcript levels (continuous variable), indicated that high SLIT2 transcript levels were independently associated with increased OS in the IC-APL cohort (hazard ratio, HR: 0.94, 95% CI: 0.92–0.97; *p* < 0.001) (Figure 1C). Similar results were observed when we analyzed APL samples from the TCGA dataset (Appendix A). Of the 80 patients who achieved complete remission (CR), 9 patients (9%) relapsed (IC-APL cohort). The 5-year disease-free survival (DFS) rate was 87% (95% CI: 80–92%). No differences were detected between patients with low and high SLIT2 levels (Appendix A).

Several studies reported epigenetic repression or activation of SLIT-ROBO in distinct tumors linked to tumor suppressive or pro-tumorigenic functions [13,14]. Hence, we decided to investigate the methylation status of the SLIT2 gene in APL patients, using the TCGA cohort. As a result, we observed a negative correlation between the global methylation status of SLIT2 and SLIT2 gene expression (r = −0.3604; *p* = 0.187, Appendix A) and a wide heterogeneity of SLIT2 methylation across APL patients (https://www.cbioportal.org/). Furthermore, we found a positive correlation between SLIT2 global methylation and WBC count (r: 0.458, *p* = 0.085; Appendix A). In accordance with our previous analysis, patients with high SLIT2 methylation and thus low transcript levels were associated with decreased OS and DFS (*p* < 0.001, Appendix A).

Next, we evaluated cellular processes that were enriched in APL patients with high and low SLIT2 expression. Clustered analysis using the transcriptome of APL patients (included in the TCGA cohort) demonstrated distinguished gene expression signature between the two groups (Figure 1D). In total, 108 genes were up-regulated and 232 genes were down-regulated in APL SLIT2^low^ patients versus SLIT2^high^ expression (Figure 1E; Appendix A). Interestingly, we found that APL patients with high SLIT2 expression were associated with cell cycle controllers (CDKN1A, TP53, and TP73), whereas patients with low SLIT2 expression were linked to cell proliferation genes (AKT1, MYB, MIK67, and MAP3K11) and poor APL/AML prognosis marker (FOXM1, AREG, and PIM1). Subsequently, gene ontology (GO) and gene set enrichment analysis (GSEA) associated SLIT2^high^ APL patients with the terms “PML nuclear body formation”, “autophagosome organization”, and “mitotic G2/M arrest”, while low SLIT2 APL patients were associated with “regulation of transcription by RNA pol II”, “cell cycle progression”, and “DNA replication” (Figure 1F,G). These results suggest that APL patients with low or absent SLIT2 expression differ in their metabolic and proliferative state compared with patients with high SLIT2 expression.

### 2.2. SLIT2 Significantly Impacts APL Cell Proliferation and Cell Cycle Progression

To address the impact of SLIT2 on APL cell proliferation, we compared the effect of *SLIT2* knockdown and SLIT2 peptide treatment on primary APL cells and APL cell lines. The proliferation rate of shSLIT2 primary APL blast cells was significantly increased from day six on compared with shCTRL cells (Figure 2A, left panel). Contrarily, these effects were reverted upon SLIT2 treatment (Figure 2A, right panel). Similar results were found when NB4 and NB4-R2 cell lines silenced for *SLIT2* or treated with SLIT2 peptide were stained for Ki-67 (Figure 2B). Short-term culture of synchronized cells showed an accumulation of shSLIT2 NB4 and NB4-R2 in the G2/M-phase, while treatment with SLIT2 peptide promoted cell cycle arrest in in the S-phase for NB4, but not NB4-R2 cells (Figure 2C). When we analyzed the ability of colony formation, peptide treated primary blasts, NB4 and NB4-R2 cells presented a significantly lower number of colonies compared with their vehicle, while SLIT2-knockdown promoted increased colony formation capacity in serial replating assays (Figure 2D). Interestingly, the SLIT2 locus was reported to encode the intronic microRNA-218, which was described to target the B lymphoma Mo-MLV insertion region 1 homolog (BMI-1) gene [15,16]. Using the TCGA cohort, we observed a strong positive correlation between miR218 and SLIT2 gene expression in APL patients (Appendix A), while the SLIT2-knockdown in APL cell lines (NB4 and NB4-R2) resulted in increased BMI-1 expression (Appendix A). In summary, SLIT2 silencing/treatment resulted in increased/decreased APL cell cycle progression and proliferation, respectively. Moreover, using the DepMap portal to evaluate the gene dependency dataset (DEMETER scores) from 17 AML cell lines included in the Cancer Cell Line Encyclopedia (CCLE), we observed an increase in cell proliferation for most of the cell lines upon SLIT2 knockdown.

### 2.3. SLIT2 Impacts on APL Cell Viability and APL Drug Induced Apoptosis

Next, we investigated whether SLIT2 promotes resistance to drug-induced apoptosis. For this purpose, we treated the silenced NB4 and NB4-R2 cell lines with ATO and ATO + ATRA for a period of 24, 48, and 72 h to measure the apoptotic rate via standard flow-cytometry procedures. As a result, shSLIT2 cells were more resistant to ATO induced apoptosis (Figure 2E), while the treatment with SLIT2 peptide promoted increased NB4 and NB4-R2 cell death (Figure 2F). Interestingly, no significant difference in apoptosis was detected in the presence of SLIT2 between the different time-points (24, 48, and 72 h), suggesting that SLIT2 induced apoptosis is triggered at the onset of treatment. Additionally, short-term treatment (24 h) with ATO and ATO + ATRA showed an accumulation of shSLIT2 APL cell lines in the G2/M-phase in comparison with control (shCTRL) cells with no difference in the Sub-G0 phase (Figure 2G).

### 2.4. Slit2 Knockdown was Associated with Increased Leukocyte Count and Decreased OS in an APL Knockin Murine Model

Finally, we assessed the impact of SLIT2 on leukemogenesis in vivo using the PML-RARα transgenic mouse model. To do so, we transplanted lethally irradiated PepBoy mice (six animals per group) with 1 × 10^6^ transduced hCG-PML-RARα blasts (with shSlit2/shCTRL) combined with 1 × 10^5^ recipient-type BM cells (Figure 3A). The median follow-up was 26 days (95% CI: 20–32 days), whereby animals transplanted with shSLIT2 APL blasts had significantly lower OS (22 days, 95% CI: 21–24 days) compared with control mice (33 days, 95% CI: 29–37 days) (*p* = 0.002, Figure 3B). To monitor APL progression, we performed weekly bleedings, which indicated a significant increase in blood cell count (*p* < 0.001, Figure 3C), emergence of anemia, and decreased platelet counts in the shSLIT2 animals at week two. Nevertheless, at the time of sacrifice, both groups (shCTRL and shSLIT2) presented similar levels of blast infiltration (defined as CD45.2+CD117+Gr1+CD34+CD16/32+ and CD34-CD16/32+) in BM, spleen, and PB (Figure 3D–F). Together, our data suggest that the downregulation of SLIT2 increases the proliferative capacity of APL blasts, leading to a more aggressive course of the disease and, consequently, decreased overall survival.

## 3. Discussion

Although the PML-RARA induced differentiation block is well established, the translocation itself is not sufficient to recapitulate the entire disease burden [14]. This suggests that APL emergence is the result of a complex interplay including in part the occurrence of driver mutations and epigenetic modifiers [17,18]. Notably, the *SLIT2* gene became of interest after the publication of Amodeo et al., demonstrating upstream functions of PML regulating the *SLIT2* expression via epigenetic repression [9].

Studies in AML non-APL reported an up-regulation of ROBO1 and 2, while all three members of the SLIT family were downregulated. However, no impact was observed between any of the *SLIT*-*ROBO* genes and the clinical outcome of patients [19]. Contrary to this study, we found that *SLIT2* transcript levels were up-regulated, while those of *SLIT1* and *ROBO1*, 3, and 4 were down-regulated in APL patients compared with healthy stem progenitor cells and promyelocytes. These discrepancies in results are not unusual between APL and non-APL AML studies, given the unique biology of APL. Interestingly, several studies suggested that the expression of *SLIT2* is downregulated by the enhancer of zeste homolog 2 (*EZH2*), a subunit of the polycomb repressor complex 2 (PRC2) [20]. Moreover, Amodeo et al. demonstrated that the loss of PML nuclear body (NB) formation in neural/stem progenitor cells (NPCs) leads to an increase of *SLIT2* expression linked to a decrease of EZH2 mediated Histone 3 lysine 27 tri-methylation (H3K27me3) [9]. In the context of APL, the up-regulation of SLIT2 could be explained by the lack of PML NB formation, but concurrently questioned by the presence of PRC2 in the PML-RARA repressive complex. Hence, whether PML-RARA can bind to the *SLIT2* promoter site and to which extent EZH2 enzymatic activity can modulate *SLIT2* transcript levels remain to be elucidated.

Further analysis depicted better overall survival and disease-free survival for APL patients with high *SLIT2* expression and a negative correlation between *SLIT2* and WBC at diagnosis. Concordantly, the methylation status of *SLIT2* positively correlated with patients’ WBC, suggesting that the downregulation of *SLIT2* is a result of epigenetic modification. In silico analysis showed a distinctive RNA signature between high and low *SLIT2* patients. A recent study reported that the distribution of Histone 3 lysine 9/14 acetylation (H3K9/K14ac) and H3K27me3 distinctively differs in high-risk patients compared with intermediate/low risk patients [18,21]. Given that the majority of *SLIT2* low patients were categorized as high-risk patients, it is possible that these disparities result from different epigenetic landscapes reflecting different risk groups. Further gene ontology and enrichment score analysis associated low *SLIT2* expression with cell cycle progression, negative regulation of apoptosis, and DNA replication, reinforcing the notion that *SLIT2* acts as a tumor suppressor gene in APL.

Of interest is the fact that, along with the transcription of *SLIT2*, the micro-RNA precursor 218 (miR-218) is transcribed [15]. The role of miR-218 in tumorigenesis has been investigated in various solid tumors such as bladder and nasopharynx cancer, suggesting tumor suppressive functions related to cell proliferation and viability [16,22]. To the best of our knowledge, only one study evaluated the role of miR-218 in APL, showing decreased expression of miR-218 in APL patients compared with healthy BM mononuclear cells. Furthermore, overexpression of miR-218 led to reduced APL cell proliferation and viability in vitro [23]. Our results demonstrate that the genetic silencing of SLIT2 leads to increased expression of BMI-1 in APL cell lines. The mechanism of action suggested in APL and other studies is that miR-218 can inhibit *BMI*-1, which leads to an increase of p16INK4A to halt the cell cycle and p19ARF to inhibit p53 degradation [24,25,26]. It is important to point out that, in the context of myeloid neoplasms, the role of BMI-1 was associated with stemness control rather than cell proliferation [27]. Other studies on gastric and breast cancer suggest that SLIT2 executes its tumor suppressive function via the regulation of the AKT/β-catenin signaling pathway [28,29]. Furthermore, a study in bladder cancer reported that miR-218 not only suppresses BMI-1, but also upregulates *PTEN*, thereby reducing the levels of phosphorylated AKT [16]. Although further confirmation is required, it is conceivable that the impact of *SLIT2* on WBC is the result of miR-218 mediated inhibition of BMI-1 and other downstream targets.

As hypothesized, *SLIT2* silencing in APL cell lines and primary APL cells led to increased cell proliferation, cell cycle progression, and resistance to ATO/ATRA drug induced apoptosis. The cell cycle analysis in APL cells upon drug-treatment suggested that shSLIT2 cells are more resistant to ATO treatment as a result of increased cell proliferation at the early time points of the treatment. It remains unclear whether the increased shSLIT2 proliferation rate upon ATO treatment is due to the downregulation of genes associated with ATO-response (TP53, TP73, and CDKN1A) or a positive selection of clones with decreased SLIT2 expression in the bulk of shSLIT2 cells. Conversely, exogenous treatment with recombinant SLIT2 protein promoted anti-tumorigenic functions. Interestingly, a more discrete anti-tumorigenic effect was observed in NB4-R2 cells (ATRA-resistant). This could be explained by the more aggressive phenotype observed on this cell line in comparison with the parental counterpart (NB4 cells) [30] Intriguingly, the knock-down of SLIT2 appeared to execute a more prominent effect than the peptide treatment, arguing that SLIT2 carries out not solely receptor mediated functions, but also important intracrine signaling worth further exploration.

Finally, *SLIT2* silenced primary murine APL cells displayed significantly lower overall survival and more rapid increase of leukocyte counts, indicating that the kinetics of APL progression are accelerated in the shSLIT2 group compared with controls. These results reinforce the hypothesis that SLIT2 primarily affects APL proliferation and cell cycle progression. Given that all animals died from full blown leukemia, no differences in APL cellularity were detected at the time of death. To capture these differences, both groups would need to be sacrificed at a set time point after transplant.

## 4. Materials and Methods

### 4.1. Patients Samples

A total of 94 consecutive patients with newly diagnosed APL, who were enrolled in the International Consortium of Acute Promyelocytic Leukemia (IC-APL) study, were included. Details about the diagnosis, classification, and treatment protocol are published elsewhere [12]. All procedures used were approved by the Ethics Committee of the Medical School of Ribeirao Preto, University of Sao Paulo, and by the National Commission of Ethics in Research, National Health Council, Ministry of Health (CONEP) (Registry# 12920; Process number: 13496/2005; CAAE: 155.0.004.000-05). Informed consent was obtained from all patients and approved by the Research Ethics Board as described in the section pertaining to ethics approval and consent to participate. All methods were carried out in accordance with the approved guidelines and to the Declaration of Helsinki. Bone marrow mononuclear cells (BM-MNCs) were isolated via Ficoll separation and cryopreserved.

### 4.2. Gene Expression Profile Using Public Datasets

To validate our gene expression data in APL patients, we analyzed the expression of all *SLIT*-*ROBO* members and their targets using the following datasets: TCGA (The Cancer Genome Atlas) study for acute myeloid leukemia [31], GSE6891 [32], the BeatAML study [33], and the BloodSpot database [11].

### 4.3. Gene Set Enrichment Analysis (GSEA) for SLIT2 Biological Pathways in APL

Gene set enrichment analysis (GSEA) was performed using the Broad Institute software (http://software.broadinstitute.org/gsea/index.jsp). All genes from the RNA-seq of the TCGA AML cohort were pre-ranked according to their differential expression (fold change) and APL patients (*n* = 16) were categorized into high and low expression of SLIT2, using their value equal to 1 as a cutoff (≤1, low SLIT2 expression; ≥1, high SLIT2 expression). Enrichment scores (ESs) were calculated based on Kolmogorov–Smirnov statistics, tested for significance using 1000 permutations, and normalized (NES) to consider the size of each gene set. A false discovery rate (FDR) cutoff of 25% (FDR q-value < 0.25) was used [34].

### 4.4. Differential Expression Analysis

Differentially expressed genes were derived using the *limma* package (lmFit function) for microarray and *DESeq2* for RNA-seq. Contrast matrices between selected groups are listed in Appendix A. Genes were considered differentially expressed if Benjamini–Hochberg FDR < 0.05.

### 4.5. Gene Expression Profile of SLIT2 in APL Patients

All samples used for gene expression analyses were obtained at diagnosis from bone marrow aspirates and processed according to standard techniques. The gene expression profile was performed as described previously elsewhere [35]. Briefly, the *SLIT2* gene expression was determined by real-time reverse transcriptase polymerase chain reaction using TaqMan Gene Expression Assay (Hs01061407_m1, Applied BioSystems, Foster City, CA, USA), following the manufacturer’s instructions. The gene expression values of SLIT2 were calculated as relative quantification using the ∆Ct method and expressing the results as 2-ΔΔCt, in which ΔΔCt = ΔCtpatients – ΔCtNB4 cell line (NB4 cell line, a human acute promyelocytic leukemia cell line, positive for PML/RARA fusion gene).

### 4.6. Cell Lines and Drugs

All cell cultures were maintained in a humidified atmosphere at 37 °C with 5% CO_2_. NB4 (all-trans retinoic acid, ATRA-sensitive) and NB4 R2 (ATRA-resistant) cell lines were kindly provided by Dr. Pier Paolo Pandolfi (Harvard Medical School, Boston, MA, USA), and maintained in RPMI 1640 (Gibco, Carlsbad, CA, USA) supplemented with 10% fetal bovine serum (FBS) (Gibco), L-glutamine (2 mM), and penicillin/streptomycin (Invitrogen, Carlsbad, CA, USA). Mycoplasma contamination was routinely tested. All leukemia cell lines were tested by short tandem repeat analysis. The HS5 (ATCC, CRL-11882), HS27A (ATCC, CRL-2496), and HEK293T (ATCC, CRL-3216) cell lines were obtained from American Type Culture Collection and grown in Dulbecco’s modified Eagle medium with 10% FBS. ATRA and arsenic trioxide (ATO) were obtained from Sigma-Aldrich (St. Louis, MO, USA) and used at a concentration of 1 µM for all in vitro assays. Cytarabine (Ara-C) was obtained from Blau Pharmaceutics (Blau Pharmaceutics, Sao Paulo, Brazil). SLIT2 peptide was obtained from PeproTech (produced in 293T cells—PeproTech EC, Ltd. London, UK) and used at the concentration of 50 ng/mL for all in vitro assays.

### 4.7. Lentiviral Vectors and Lentivirus Production

Recombinant lentivirus encoding the short hairpin sequence (shRNA) targeting the human and murine *SLIT2* gene (TRCN0000363636 (human) and TRCN 0000328111 (mouse)) (hereafter called shSLIT2 and shSlit2 for the human and mouse constructs, respectively) were generated using a MISSION pLKO.1 lentivector (Sigma-Aldrich) in HEK293T cells according to the three-plasmid packaging procedure described elsewhere [36]. The MISSION-shRNA sequence selection was performed using the Broad Institute RNAi consortium data bank [37]. Lentiviral particles were used to infect APL cell lines (NB4 and NB4 R2; Multiplicity of Infection [MOI] = 6) and primary (human and murine; MOI > 50) APL blast samples using Retronectin-coated plates (Takara, Kusatsu, Shiga, Japan) in the presence of 8 µg/mL polybrene (Sigma-Aldrich) overnight. Transduced cell lines were selected with puromycin (Sigma-Aldrich) at the dose of 0.5 µg/mL, for 3–5 passages (~10 days), and posteriorly used for in vitro assays. The efficiency of infection was further confirmed by gene expression quantification (reduction of at least 70% of expression). An shRNA sequence that does not target human and murine genes (referred to as scrambled) was used as a control (shCTRL).

### 4.8. In Vitro Assays

*Cell proliferation:* NB4 and NB4-R2 cells were treated with SLIT2 peptide (50 ng/mL) for 24, 48, and 72 h. For *SLIT2*-silenced APL cell lines, cells were treated twice with thymidine (2 mM) for 18 h to induce cell cycle arrest at the G1/S boundary. Cells were subsequently seeded in six-well plates and 1 million cells were collected and fixed with 70% ethanol at distinct timepoints: 24, 48, and 72 h, and stored at −20 °C. For both conditions (SLIT2 treatment and shSLIT2 cells), Ki-67 staining was performed following the manufacturer’s instructions (Ki-67 PE clone SolA15; BioLegend, San Diego, CA, USA). Next, the mean of fluorescence intensity (MFI) was obtained by flow cytometry standard techniques using a FACSCantoII instrument (Becton-Dickison, Franklin Lakes, NJ, USA). IgG isotype was used as a negative control for each condition. In parallel, cells were seeded at a density of 1 × 10^4^ cells/mL in 10 cm dishes and the cell number was counted daily for seven days.

*Cell cycle analysis:* Cell cycle phases were determined by BD CycletestTM Plus DNA Reagent Kit (Becton-Dickinson, Mountain View, CA, USA) according to the manufacturer’s instructions. A total of 4 × 10^5^ synchronized shSLIT2 and shCTRL cells in G1/S phase were seeded in 24-well plates and collected and fixed at distinct timepoints: 24, 48, and 72 h. In addition, two experimental setups were performed: (1) NB4 and NB4-R2 cells treated with SLIT2 peptide for 24, 48, and 72 h were collected and fixed for further evaluation; (2) transduced APL cell lines (shSLIT2 and control) were treated with vehicle (NaOH plus Dimethyl sulfoxide [DMSO] 0.001%), ATO (1 µM), and ATO + ATRA (1 µM each) for 24 h, and then collected and fixed for further evaluation. DNA content distribution was acquired with the FACSCalibur cytometer (Becton-Dickinson) and analyzed using the FlowJo software (Treestar, Inc., San Carlos, CA, USA).

*Apoptosis assay:* A total of 5 × 10^5^ transduced cells were seeded in 24-well plates and incubated in complete medium for 24, 48, and 72 h in the presence of vehicle (for ATRA, DMSO; for ATO, NaOH; and, for cytarabine (Ara-C), saline-buffer; both at a final concentration of less than 0.003% by volume), ATO (1 µM, alone or in combination with 1µM of ATRA), and cytarabine (Ara-C, 10 nM). NB4 and NB4 R2 cells were seeded using the same settings as transduced cells and were treated with SLIT2 peptide for 24, 48, and 72 h. The apoptosis rate was determined using the Annexin V-APC and propidium iodide (PI) binding assay (BD Biosciences, San Jose, CA, USA). All specimens were acquired by flow cytometry (FACSCantoII; Becton-Dickison) and analyzed with the FlowJo software (Treestar, Inc.). All experiments were performed in triplicate and, for each sample, a minimum of 10,000 events were acquired.

*Colony formation assay:* Colony formation capacity in APL cell lines (submitted to SLIT2 treatment and transduced with shCTRL/shSLIT2) was evaluated in semisolid methylcellulose medium in four consecutive replatings (1.5 × 10^3^ cells/mL; MethoCult 4230; StemCell Technologies Inc., Vancouver, BC, Canada). For primary APL blasts, semisolid methylcellulose medium supplemented with cytokines was used (1 × 10^4^ cells/mL; MethoCult H4434; StemCell Technologies Inc., Vancouver, BC, Canada). Additionally, primary APL blasts and APL cell lines were treated with SLIT2 peptide and submitted to clonogenic evaluation. Colonies were detected after 10–14 days of culture by adding 1 mg/mL of MTT (3-(4,5-dimethylthiazol-2-yl)-2,5-diphenyl tetrazolium bromide) reagent and scored with the Image J quantification software (US National Institutes of Health, BethesdaMD, USA).

### 4.9. Quantitative PCR

Total RNA from transduced (shSLIT2 and control) APL cell lines (NB4 and NB4-R2) and murine hCG-PML-RARA blasts was obtained using TRIzol reagent (Thermo Fisher Scientific, Carlsbad, CA, USA). The cDNA was synthesized from 1 µg of RNA using High-Capacity cDNA Reverse Transcription Kit (Thermo Fisher Scientific). Quantitative PCR (qPCR) was performed with an ABI 7500 Sequence Detector System (Life Technologies, Carlsbad, CA, USA) with the human and murine *GAPDH*, *RPL30*, and *ACTB* Standard Kit as endogenous controls. The murine *Slit2* and the human BMI1 gene expression were determined by real-time reverse transcriptase polymerase chain reaction using TaqMan Gene Expression Assay: *Slit2*, Mm01216521_m1; *BMI1*, Hs00995519_g1 (Applied BioSystems, Foster City, CA, USA), following the manufacturer’s instructions. The relative quantification value was calculated using the equation 2 − ΔΔCT. A negative “no template control” was included for each probe evaluated.

### 4.10. Generation of PML-RARA Knockin-Induced APL Mice by Bone Marrow Transplantation

For APL phenotype induction, 1 × 10^6^ transduced leukemic blast cells from hCG-PML-RARA mice (expressing the CD45.2) [36] were transplanted into lethally irradiated eight-week-old Pep boy mice (B6.SJL-Ptprca Pepcb/BoyJ, The Jackson Laboratory). After 2 weeks, chimerism was evaluated by CD45.1 (Pep boy, provided by Jackson Laboratory – B6.SJL-Ptprca Pepcb/BoyJ) and CD45.2 markers (Becton–Dickinson) by flow cytometry in peripheral blood. Mice were followed for overall survival analysis and sacrificed when tumor engraftment reached ethical limits (>90% of engraftment) or mice showed severe signs of illness. At the end of the experiment, animals were harvested and subjected to analysis of peripheral blood, spleen, bone marrow, and hematological parameters. Viable early and late promyelocytes in the spleen and bone marrow were evaluated by analysis of CD34 and CD16/32 (CD34+CD16/32+, Early Pro/CD34-CD16/32+, Late Pro) inside the population of lineage negative cells (lineage markers defined by CD3e, CD19, B220, Ter119, NK1.1, CD4, and CD8) positive for CD117 and Gr1intermediate [38] (Becton–Dickinson) by flow cytometry. All animals were housed under specific pathogen-free conditions in individually ventilated cages during the whole experiment and were maintained according to the Guide for Care and Use of Laboratory Animals of the National Research Council, USA, and to the National Council of Animal Experiment Control recommendations. All experiments were approved by the Animal Ethics Committee of the University of Sao Paulo (#095/2018).

### 4.11. Statistical Analysis

Patient baseline characteristics were reported descriptively. Fisher’s exact test or Chi-square test, as appropriate, was used to compare categorical variables, and Kruskal–Wallis test was used to compare continuous variables. The cut-off used to dichotomize APL patients into two groups (i.e., low expression, <1.53; high expression, ≥1.53) was set by simulating all possibilities [39], followed by the inspection of Kaplan–Meier (KM) curves.

Overall survival was defined as the time from diagnosis to death from any cause; those alive or lost to follow-up were censored at the date last known alive. Early mortality was defined as death occurring within 30 days from diagnosis. For patients who achieved CR, DFS was defined as the time from CR achievement to the first adverse event; that is, relapse or death from any cause, whichever occurred first. Patients who were alive without disease relapse or secondary malignancy were censored at the time they were last seen alive and disease-free. To adjust *SLIT2* expression for confounding factors that were determined through stepwise selection, we performed a multivariate Cox proportional hazard regression. All *p*-values were two-sided with a significance level of 0.05. All calculations were performed using Stata Statistic/Data Analysis version 14.1 (Stata Corporation, College Station, TX, USA), statistical package for the social sciences (SPSS) 19.0 and R 3.3.2 (The CRAN project, www.r-project.org) software.

## 5. Conclusions

In summary, low expression of *SLIT2* is a poor prognostic marker in APL, which negatively correlates with leukocytes counts. The downregulation of *SLIT2* appears to be the result of epigenetic repression, while the definite mechanism is still unclear. Interestingly, our data suggest that the function of SLIT2 is not only receptor mediated, but also intracellularly, affecting APL cell proliferation in vitro and in vivo.

## Figures and Tables

**Figure 1 cancers-12-03134-f001:**
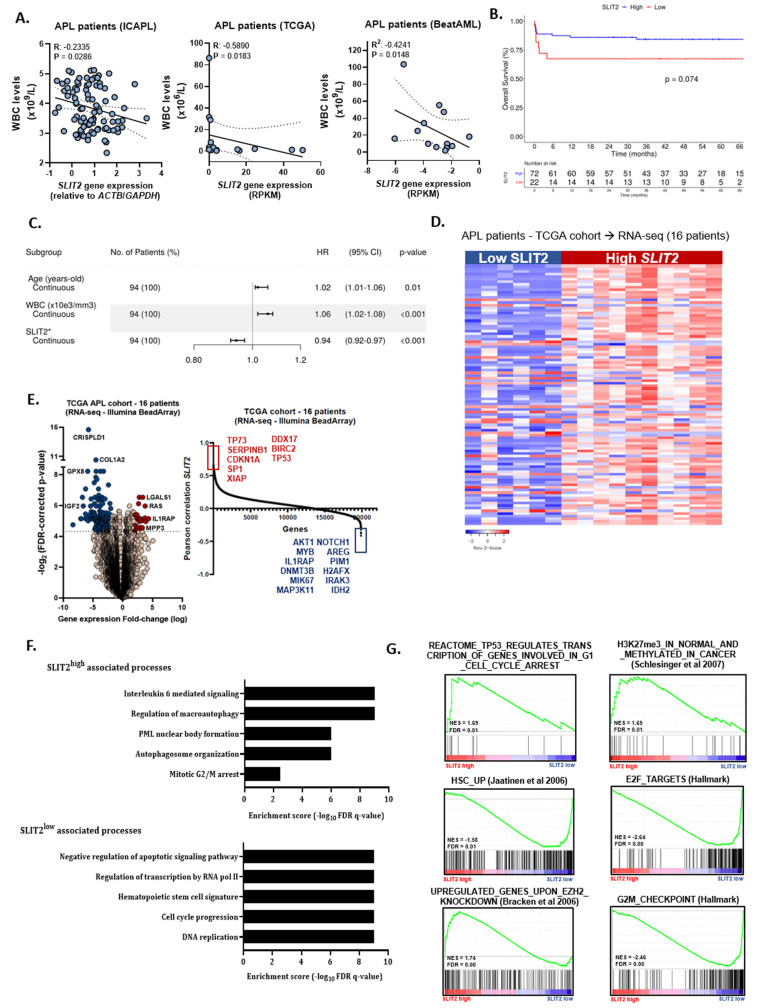
Clinical role of SLIT2 in acute promyelocytic leukemia (APL). (**A**) Correlation analysis of *SLIT2* expression and white blood cell count in APL patients included in the International Consortium of Acute Promyelocytic Leukemia (IC-APL) cohort (left panel), in the TCGA cohort (middle panel) and in the BeatAML study (right panel). The r and *p*-values are indicated on the figure, Spearman correlation test. (**B**) The probability of overall survival in APL patients according to the *SLIT2* expression. Survival curves were estimated using the Kaplan–Meier method and the log-rank test was used for comparison. (**C**) Multivariate Cox model for overall survival. Differential expression of *SLIT2* can categorize APL patients into different risk groups. (**D**) Heatmap of differentially expressed genes (supervised clustering) in APL samples from TCGA cohort dichotomized based on the *SLIT2* gene expression (dichotomization point: expression below 1; groups: high (*n* = 10) and low (*n* = 6) expression). (**E**) Volcano plot comparing APL patients with low *SLIT2* expression versus APL patients with high *SLIT2*. Fold change was set at 2 for upregulated and downregulated gene expression. Significance was set at a corrected FDR < 0.05. A total of 232 genes were downregulated (blue) and 108 were upregulated (red). (**F**) Gene ontology (GO) and (**G**) gene set enrichment analysis (GSEA) on a ranked gene list based on the leading-edge genes for *SLIT2* expression in 16 de novo APL patient samples from TCGA study. Genes were ranked based on Pearson correlation using the *SLIT2* gene expression. Normalized enrichment score (NES) and false discovery rate (FDR) was used to determine significance.

**Figure 2 cancers-12-03134-f002:**
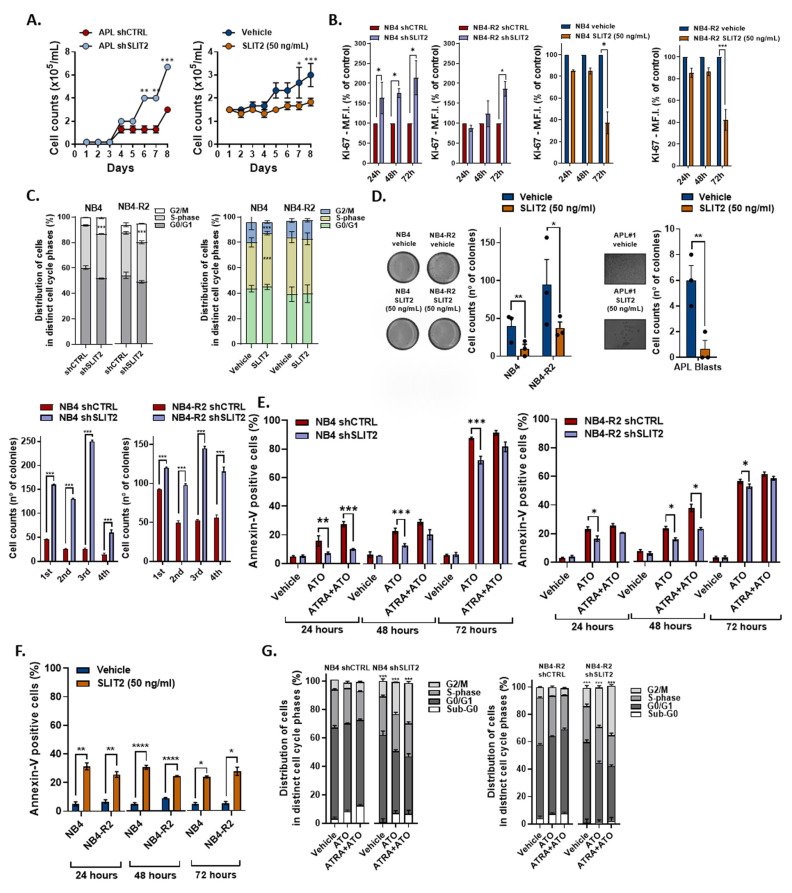
Effect of SLIT2 on APL cell proliferation. (**A**) Growth curves using human primary blasts from APL patients, transduced with shSLIT2 or scrambled control (shCTRL) (left panel), and treated with SLIT2 peptide (50 ng/mL) (right panel). (**B**) Ki67 staining and (**C**) cell cycle analysis of NB4 and NB4-R2 cell lines transduced with shSLIT2 or scrambled (shCTRL) and wild-type NB4 and NB4-R2 cells treated with SLIT2 peptide (50 ng/mL). (**D**) Representative example of one of three independent experiments of colony formation assay in methylcellulose using wild-type NB4, NB4-R2 (left panel), and primary human APL blast cells (right panel) treated with SLIT2 peptide (50 ng/mL). Graphic bars represent the average of the number of colony-forming cells when treated. Lower panel shows the graphic bars representing the average of the number of colonies from APL cell lines (NB4 and NB4-R2) transduced with shSLIT2 and control (shCTRL). Effect of SLIT2 on drug-induced apoptosis. Percentage of apoptotic transduced cells (**E**) and SLIT2-treated cells (**F**) after 24, 48, and 72 h in culture with apoptotic stimulus. (**G**) Cell cycle analysis of NB4 and NB4-R2 cell lines transduced with shSLIT2 or scrambled (shCTRL) upon arsenic trioxide (ATO) (1 µM) and ATO + all-trans retinoic acid (ATRA) (1 µM each), for 24 h. Graphic bars represent the mean and standard error (SD) of four independent experiments. * indicates *p* < 0.05, ** indicates *p* < 0.01, *** indicates *p* < 0.001, **** indicates *p* < 0.0001.

**Figure 3 cancers-12-03134-f003:**
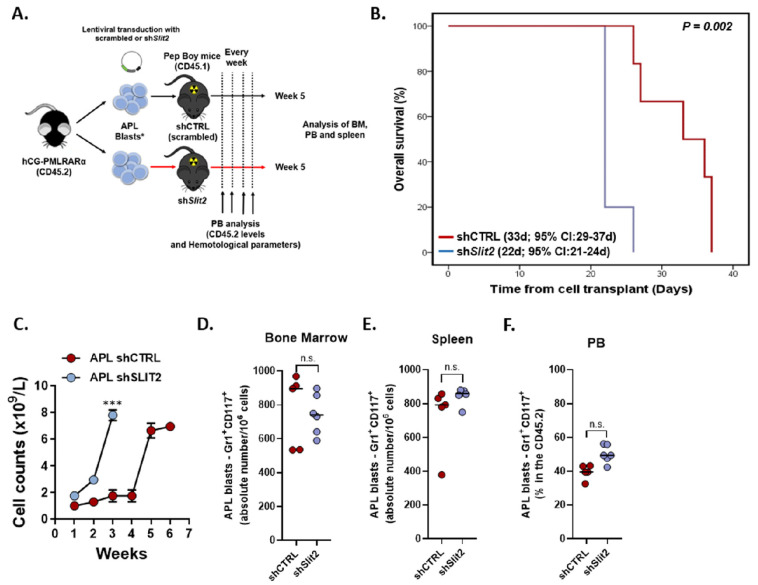
*SLIT2* knockdown induces APL clonal expansion and reduces overall survival in vivo. Overview of the mouse model for APL generation. (**A**) Schematic representation of the generation of the isogeneic mouse model for APL engraftment using blasts from the hCG-PMLRARA mice (CD45.2) transplanted in Pepboy mice (CD45.1). (**B**) The probability of overall survival in mice transplanted with murine APL blasts transduced with shSlit2 and scrambled as a control (shCTRL). Survival curves were estimated using the Kaplan–Meier method and the log-rank test was used for comparison. (**C**) Weekly bleedings of mice were used to determine the leukocyte count (*10^9^/L) and monitor disease progression. At sacrifice, APL blasts and cells from Pepboy mice were analyzed by flow cytometry using markers against CD117, Gr1, CD16/32, and CD34, as indicated (inside the population CD45.2+ and lineage negative). Scatter plots showing engraftment of donor APL blast cells (**D**) in bone marrow (BM), (**E**) spleen (percentual of engraftment and spleen weight), and (**F**) peripheral blood (PB) of transplanted mice at sacrifice. Data were expressed as median values. *** indicates *p* < 0.001; n.s. indicates not significant.

**Table 1 cancers-12-03134-t001:** Baseline characteristics.

Characteristic	Characteristic	All Patients		*SLIT2* Expression				*p* Value
				Low Expression		High Expression		
		No.	%	No.	%	No.	%	
Gender	Female	47	50	14	60.9	33	46.5	0.337
	Male	47	50	9	39.1	38	53.5	
	Age, median	35.7		39.4		33.6		0.482
	(range)	(18.9, 73.6)		(19, 65.43)		(18.9, 73.6)		
ECOG performance status	0	59	62.8	11	47.8	48	67.6	0.243
	1	14	14.9	6	26.1	8	11.3	
	2	12	12.8	4	17.4	8	11.3	
	≥3	9	9.6	2	8.7	7	9.9	
Relapse-risk group	Unknown	-	-	-	-	-	-	
	Low risk	16	17	2	8.7	14	19.7	0.307
	Intermediate risk	40	42.6	9	39.1	31	43.7	
	High risk	38	40.4	12	52.2	26	36.6	
*FLT3*-mutational status	Mutated	14	19.7	5	29.4	9	16.7	0.299
	Non-mutated	57	80.3	12	70.6	45	83.3	
	Unknown	23	-	6	-	17	-	
*PML* breakpoint	BCR1	55	65.5	10	47.6	45	71.4	0.061
	BCR2	2	2.4	-	-	2	3.2	
	BCR3	27	32.1	11	52.4	16	25.4	
	Unknown	10	-	2	-	8	-	
	WBC counts (×10^9^/L), median	6.78		15.02		3.3		0.024 *
	(range)	(0.22, 132.5)		(0.89, 126.8)		(0.22, 132.5)		
	Platelet counts (×10^9^/L), median	26.7		28		24		0.329
	(range)	(4, 128)		(4, 92)		(4, 128)		
	Hemoglobin (g/dL), median	8.8		8.6		8.8		0.898
	(range)	(3.4, 14.1)		(4.7, 13.3)		(3.4, 14.1)		
	Creatinine (mg/dL), median	0.8		0.9		0.8		0.738
	(range)	(0.42, 2.2)		(0.42, 1.88)		(0.5, 2.2)		
	Uric acid (mg/dL), median	3.9		4.2		3.7		0.099
	(range)	(1.1, 9)		(2.5, 7.1)		(1.1, 9)		
	Fibrinogen (mg/dL), median	159.5		148		165		0.082
	(range)	(10, 605)		(48, 271)		(10, 605)		
	Albumin, g/dL	4		4.5		4		0.115
	(range)	(2.2, 5.3)		(2.9, 5.2)		(2.2, 5.3)		

Abbreviations: ECOG, Eastern Cooperative Oncology Group; FLT3, Fms-like tyrosine kinase 3; BCR, breakpoint cluster region; WBC, white blood cell. NOTE: * indicates statistically significant differences. Missing values were excluded.

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
