# Peer review of "Reduced SLIT2 is Associated with Increased Cell Proliferation and Arsenic Trioxide Resistance in Acute Promyelocytic Leukemia"

_cancers, 2020, doi:10.3390/cancers12113134_

Round 1
Reviewer 1 Report
The authors have provided sufficient information in the revised manuscript. However, supplementary table S1 is still empty in the Zip file, although this may be a problem with my computer. Please check if the submitted file is OK.
Reviewer 2 Report
The authors addressed my concerns.
This manuscript is a resubmission of an earlier submission. The following is a list of the peer review reports and author responses from that submission.
Round 1
Reviewer 1 Report
The main points to be clarified are as follows:
1. The authors stated that blasts with SLIT2high transcript levels were associated with cell cycle arrest, while SLITlow APL blasts displayed a more stem-cell like phenotype. However, in the manuscript, there is no evidence to show relationship between SLITlow and stem-cell phenotype.
2. Studies in AML non-APL reported up-regulation of ROBO1 and 2, but down-regulation of three SLITs. From the BloodSpot database, compared with healthy HSPCs and promyelocytes, the expression of SLIT2 in APL patients increased, while the expression of SLIT1, ROBO1, 3, and 4 decreased. The authors suggested that this difference may be due to the unique biology of APL, which is characterized by PML-RARA translocation. Since PML can down-regulate the expression of SLIT2 via EZH2-mediated H3K27me3, the PML-RARA translocation of APL patients may lose PML function, thereby up-regulating the expression of SLIT2. The Primary blasts, APL cell lines, and animal model showed that SLIT2 plays a tumor suppressor role in APL. How to explain that SLIT2 plays a tumor suppressor role in APL, but its expression is up-regulated in APL patients?
3. Line 144-148: Fig 2B shows that shSLIT2 increased the cell proliferation of NB4 and NB4-R2, while SLIT2 treatment decreased the cell proliferation of both cells. In Fig. 2C, shSLIT2 accumulates cells in G2/M phase in both NB4 and NB4-R2 cells. SLIT2 treatment increased the S phase population and decreased G2/M phase population in NB4 cells, however there is no effect in cell cycle distribution in NB4-R2 upon SLIT2 treatment. Why is that?
4. As authors raised that miR-218 is transcribed along with the transcript of SLIT2. However, SLIT2 is upregulated in APL while miR-218 is down-regulated. Therefore, it is necessary to reconsider the discussion in Line 267 “it is conceivable that the impact of SLIT2 on WBC is the result of miR-218 mediated inhibition of BMI-1 and other downstream targets.”
5. Please comment on how SLIT2 promotes cell death induced by ATO and ATRA. Are there any genes related to this phenomenon in gene expression analysis?
6. Line 131: the supplementary table S1 is missing from the PDF.
7. Line 121: What is the correlation between the methylation status of SLIT2 and SLIT2 transcript level in patients?
8. Line 116-118, it is not clear which group of patients was discussed.
9. Line 135-138: The label in Figure 1F has poor resolution and cannot be read. Fig 1G needs more explanation.
Author Response
RESPONSE TO REVIEWERS
REVIEWER #1: COMMENTS FOR THE AUTHOR
The main points to be clarified are as follows:
- The authors stated that blasts with SLIT2high transcript levels were associated with cell cycle arrest, while SLITlow APL blasts displayed a more stem-cell like phenotype. However, in the manuscript, there is no evidence to show relationship between SLITlow and stem-cell phenotype.
RESPONSE: We thank the reviewer for the comment. In response, we include in the present version a serially plated clonogenic assay using APL cell lines transduced with shSLIT2 and the respective control (Figure 2D, last panel). Of note, the phenotypes associated with a more stem-cell like features were observed in an in-silico analysis using the APL patients from the TCGA cohort.
- Studies in AML non-APL reported up-regulation of ROBO1 and 2, but down-regulation of three SLITs. From the BloodSpot database, compared with healthy HSPCs and promyelocytes, the expression of SLIT2 in APL patients increased, while the expression of SLIT1, ROBO1, 3, and 4 decreased. The authors suggested that this difference may be due to the unique biology of APL, which is characterized by PML-RARA translocation. Since PML can down-regulate the expression of SLIT2 via EZH2-mediated H3K27me3, the PML-RARA translocation of APL patients may lose PML function, thereby up-regulating the expression of SLIT2. The Primary blasts, APL cell lines, and animal model showed that SLIT2 plays a tumor suppressor role in APL. How to explain that SLIT2 plays a tumor suppressor role in APL, but its expression is up-regulated in APL patients?
RESPONSE: We thank the reviewer for the comments. The current manuscript explores how the reduced levels of SLIT2 in APL cells can be associated with decreased overall survival in the clinical setting, and how the in vitro knockdown promotes increased cell proliferation and survival using APL cell lines as a model. In our study two analysis can be taken into consideration but should be interpreted with care. As stated by the reviewer, SLIT2 is upregulated in APL patients compared to healthy donors. This can be explained by the fact that PML-null cells induced increased SLIT2 expression and therefore patients carrying the PML-RARA translocation, which leads to PML loss would be expected to present the same phenotype (Cell Rep. 2017, 20, 411–426,doi:10.1016/j.celrep.2017.06.047.). Yet, when APL patients are dichotomized into high and low expression, the impact of SLIT2 on APL progression becomes more evident. Although SLIT2 is upregulated in APL compared to healthy cells, we suggest that the dichotomization of APL patients is, in this case, more insightful when SLIT2 is used as a prognostic marker. Moreover, we need to consider that most of the publicly available Microarray/RNA-sequence datasets are generated by bulk sequencing strategy (usually mononuclear cells) and not by sorting the leukemic blast cells specifically. Therefore, it is conceivable that the up-regulation of SLIT2 in APL patients could also arise from the stromal compartment and not from the leukemic cell. Obviously, further studies would be required to validate this statement. Of note, this peace of information were properly included in the present version of the manuscript (Discussion section, page 9, first three paragraphs).
- Line 144-148: Fig 2B shows that shSLIT2 increased the cell proliferation of NB4 and NB4-R2, while SLIT2 treatment decreased the cell proliferation of both cells. In Fig. 2C, shSLIT2 accumulates cells in G2/M phase in both NB4 and NB4-R2 cells. SLIT2 treatment increased the S phase population and decreased G2/M phase population in NB4 cells, however there is no effect in cell cycle distribution in NB4-R2 upon SLIT2 treatment. Why is that?
RESPONSE: We thank the reviewer for this observation. In fact, in our previous studies we also observed discrepancies between NB4 and NB4-R2 cell regarding the response to different drugs. The NB4-R2 cell line carries a mutated PML/RARA (L900P) transcript (Duprez, E., Leukemia, 1992, 6, 1281–1287; Duprez, E., Leukemia, 2000 14, 255–261 and Kitamura K et al., Leukemia 1997, 11: 1950-1956) and it was previously reported (Walsby E, et al. Br J Haematol. 2007 Oct; 139(1):94-7) that the expression of the FUS gene modifies ATRA response in NB4-R2. Hence, although NB4-R2 cells derive from NB4 cells, they present distinct phenotypic and functional features compared to their parental cell line. (Pereira-Martins et al., Blood 2018; 132:1532). Taking this data together it appears that NB4-R2 cells are more resistant to the SLIT2 peptide treatment (an exogeneous stimulus), while the knockdown of SLIT2 resulted in a similar phenotype in NB4 and NB4-R2 cells. Of note, this information was properly included in the present version of the manuscript (Discussion section, page 10, paragraph 1).
- As authors raised that miR-218 is transcribed along with the transcript of SLIT2. However, SLIT2 is upregulated in APL while miR-218 is down-regulated. Therefore, it is necessary to reconsider the discussion in Line 267 “it is conceivable that the impact of SLIT2 on WBC is the result of miR-218 mediated inhibition of BMI-1 and other downstream targets.”
RESPONSE: We appreciate the reviewer’s concern, and we are aware of the described downregulation of miR-218 in APL patients compared to BM mononuclear cells of healthy donors (Wang et al., Oncol Lett. 2017 Dec; 14(6): 8078–8083). However, our observation of increased SLIT2 expression coinciding with an increase in miR-218 in APL patients contrasts these findings and arises from two different cohorts. We feel that it is important that for a direct expression comparison of these two targets should be compared in the same cohort rather than between different studies. To address this point in a best possible way we used two strategies. First, we used the TCGA cohort, to compare the expression of miR-218 and SLIT2 gene expression in APL patients, which showed strong positive correlation between SLIT2 and hsa-mir-218-1 and hsa-mir-218-2 (hsa-mir-218-1 – r = 0.897; P = 0.001 and hsa-mir-218-2 – r = 0.926; P = 0.001), see supplemental figure 1H and 1I. Second, we evaluated the expression of BMI-1 in APL cell lines transduced with shSLIT2 and the scrambled control. As we can see in the supplementary figure 1J of the current manuscript, SLIT2 knockdown resulted in increased BMI-1 expression in both APL cell lines (NB4 and NB4-R2). We include a sentence in the results and discussion section highlighting this specific point together with the aforementioned sentence (page 4 and 9, 1 and 4 paragraphs, respectively, sentences in blue). Importantly, the main conclusion remained unchanged.
- Please comment on how SLIT2 promotes cell death induced by ATO and ATRA. Are there any genes related to this phenomenon in gene expression analysis?
RESPONSE: That is an excellent comment, which helped us to improve the quality of our manuscript. In fact, as we demonstrated in Figure 2E the knockdown of SLIT2 was associated with increased resistance to ATO alone or in combination with ATRA, when we evaluated drug-induced apoptosis. In the current version (please see the response to reviewer#2), we showed that this phenotype might be the result of increased shSLIT2 cell proliferation (current Figure 2G) even in the presence of ATO alone or combined with ATRA. Additionally, low levels of SLIT2 in APL patients led to decreased levels of TP53, TP73, CDKN1A and SERPINB1 (Figure 1E), which are well described targets associated with the ATO and ATRA induced apoptosis mechanism (Ablain et al., Nat Med. 2014 Feb;20(2):167-74. doi: 10.1038/nm.3441. Epub 2014 Jan 12.; Ortiz et al., Blood (2019) 134 (Supplement_1): 2719. https://doi.org/10.1182/blood-2019-131211). This suggests that cells with low SLIT2 levels could acquire an inherent resistance to drug-induced apoptosis. We include a sentence at the end of the discussion to highlight this specific point as a limitation of the study (page 10, paragraph 1, sentence in blue).
- Line 131: the supplementary table S1 is missing from the PDF.
RESPONSE: We apologize for this mistake. As requested, the aforementioned supplementary table S1 was included in the PDF and as an Excel (xls) file.
- Line 121: What is the correlation between the methylation status of SLIT2 and SLIT2 transcript level in patients?
RESPONSE: As requested, we provided in the current version of the manuscript the correlation between the methylation status of SLIT2 and SLIT2 transcript levels (Supplementary Figure 1G). Although not significant, a negative correlation was observed (r = -0.3604; P = 0.187).
- Line 116-118, it is not clear which group of patients was discussed.
RESPONSE: We apologize for the unclear information. As requested, we properly stated that the patients analyzed in this part were the patients from the ICAPL2006 study (page 3, paragraph 3, sentence in blue).
- Line 135-138: The label in Figure 1F has poor resolution and cannot be read. Fig 1G needs more explanation.
RESPONSE: We thank the reviewer for this observation and apologize for the poor resolution and limited information. Since the journal requires all figures to fit in a doc format, the quality of the figures with small details unfortunately decreased. In the present version of the manuscript, we provide the new figures in high quality (300 dpi; dots per inch) suitable for publication. Regarding the description of Figure 1G, the gene set enrichment analysis shows that APL patients with low SLIT2 expression presented a transcriptional signature that correlates with a more proliferative phenotype and decreased sensitivity to TP53 regulation. This information was included in the revised version of the manuscript (page 3, paragraph 5, sentence in blue).

Reviewer 2 Report
In their manuscript Weinhäuser et al., investigate the role of SLIT2 in APL by assessing its biological 
 activity in vitro and in vivo. The results indicated that blasts with SLIT2high transcript levels were 
 associated with cell cycle arrest, while SLIT2low APL blasts displayed a more stem-cell like 
 phenotype. Moreover, in an analysis using a cohort of patients treated with ATRA and 
 anthracyclines, high SLIT2 expression was correlated with reduced leukocyte count and improved overall survival. The authors found that SLIT2-knockdown in primary APL blasts and cell lines 
 led to increased cell proliferation and resistance to arsenic trioxide induced apoptosis. Additioanly, in 
 vivo transplant of Slit2-silenced primary APL blasts promoted increased leukocyte count
 and decreased overall survival compared to control. In general, the data presented by the authors indicate the 
 tumor suppressive function of SLIT2 in APL. 
 The manuscript is interesting and address a poor described role of SLIT2 in APL. Data are clearly presented supporting the authors’ suggestions.
I have only one question that authors should address. In particular, in fig. 2 Weinhäuser, et al., investigated whether SLIT2 promotes resistance to drug-induced apoptosis. For this 
 purpose, slit2 silenced cells were treated with ATO and ATO+ATRA. It should be interesting if the author can provide a cell cycle profile of cells treated as in Figure 2E.
Author Response
REVIEWER #2: COMMENTS FOR THE AUTHOR
In their manuscript Weinhäuser et al., investigate the role of SLIT2 in APL by assessing its biological activity in vitro and in vivo. The results indicated that blasts with SLIT2high transcript levels were associated with cell cycle arrest, while SLIT2low APL blasts displayed a more stem-cell like phenotype. Moreover, in an analysis using a cohort of patients treated with ATRA and anthracyclines, high SLIT2 expression was correlated with reduced leukocyte count and improved overall survival. The authors found that SLIT2-knockdown in primary APL blasts and cell lines led to increased cell proliferation and resistance to arsenic trioxide induced apoptosis. Additionally, in vivo transplant of Slit2-silenced primary APL blasts promoted increased leukocyte count and decreased overall survival compared to control. In general, the data presented by the authors indicate the tumor suppressive function of SLIT2 in APL. The manuscript is interesting and address a poor described role of SLIT2 in APL. Data are clearly presented supporting the authors’ suggestions.
I have only one question that authors should address. In particular, in fig. 2 Weinhäuser, et al., investigated whether SLIT2 promotes resistance to drug-induced apoptosis. For this purpose, slit2 silenced cells were treated with ATO and ATO+ATRA. It should be interesting if the author can provide a cell cycle profile of cells treated as in Figure 2E.
RESPONSE: We appreciate the reviewer’s comment, which helped us to improve the quality of our manuscript. As requested, we provide in the current version a cell cycle analysis of NB4 and NB4-R2 cells transduced with shSLIT2 and the control (shScrambled) treated with ATO and ATO+ATRA (1 µM each) (Current version - Figure 2G). Since the apoptosis levels reached over 40% in the APL cells treated with ATO+ATRA for 48 hours, we decided to evaluate the cell cycle after 24 hours of exposure, to see the differences regarding the initial behavior of SLIT2-knockdown cells compared to the control. As we can observe, shSLIT2 APL cell lines, accumulated in the S-phase and G2/M-phase in comparison with the control, upon treatment with ATO alone or in combination with ATRA. Considering that shSLIT2 promotes increased ATO resistance and cell proliferation, it is conceivable that the treatment with ATO reinforces the selection of shSLIT2 versus non-efficiently transduced cells and as a result promotes accumulation of shSLIT2 cells in S/G2 phase.
